# Molecular Mechanisms That Define Redox Balance Function in Pathogen-Host Interactions—Is There a Role for Dietary Bioactive Polyphenols?

**DOI:** 10.3390/ijms20246222

**Published:** 2019-12-10

**Authors:** Kaiwen Mu, Danni Wang, David D. Kitts

**Affiliations:** 1Food Nutrition and Health, Faculty of Land and Food Systems, The University of British Columbia, Vancouver, BC V6T-1Z4, Canada; muk@mail.ubc.ca; 2Laboratory of Bacterial Pathogenesis, Department of Microbiology and Immunology, Institutes of Medical Sciences, Shanghai Jiao Tong University School of Medicine, Shanghai 200025, China; wangdanni@shsmu.edu.cn

**Keywords:** reactive oxygen species, redox biology, catalase, pathogen infection, immune recognition, polyphenols

## Abstract

To ensure a functional immune system, the mammalian host must detect and respond to the presence of pathogenic bacteria during infection. This is accomplished in part by generating reactive oxygen species (ROS) that target invading bacteria; a process that is facilitated by NADPH oxidase upregulation. Thus, bacterial pathogens must overcome the oxidative burst produced by the host innate immune cells in order to survive and proliferate. In this way, pathogenic bacteria develop virulence, which is related to the affinity to secrete effector proteins against host ROS in order to facilitate microbial survival in the host cell. These effectors scavenge the host generated ROS directly, or alternatively, manipulate host cell signaling mechanisms designed to benefit pathogen survival. The redox-balance of the host is important for the regulation of cell signaling activities that include mitogen-activated protein kinase (MAPK), p21-activated kinase (PAK), phosphatidylinositol 3-kinase (PI3K)/Akt, and nuclear factor κB (NF-κB) pathways. An understanding of the function of pathogenic effectors to divert host cell signaling is important to ascertain the mechanisms underlying pathogen virulence and the eventual host–pathogen relationship. Herein, we examine the effectors produced by the microbial secretion system, placing emphasis on how they target molecular signaling mechanisms involved in a host immune response. Moreover, we discuss the potential impact of bioactive polyphenols in modulating these molecular interactions that will ultimately influence pathogen virulence.

## 1. Introduction

An innate immune system will respond to a pathogenic bacterium-induced mammalian infection. The host’s cellular redox balance is an important component of this response and affects the host–pathogen relationship [1,2,3]. Host intracellular killing mechanisms include the production of reactive oxygen species (ROS) generated by a respiratory burst response triggered by infectious agents [4,5]. To survive and successfully enter a host cell, bacteria must release specific protein(s) and virulence factors that function to overcome host cellular signaling defense mechanisms that affect the immune response. Virulence factors refer to a toxin, or a protein that is released by the pathogen to facilitate infecting the host [6]. Pathogens utilize diverse mechanisms to escape or counter the attack from the host immune system by disrupting immune recognition and inhibiting inflammatory responses [6].

Bacterial resistance to intracellular killing involves scavenging free radicals that are triggered by the host macrophage. Studies have shown that protein effectors produced from bacteria under oxidative stress will assist in bacteria survival [7]. Pathogens secrete a variety of effector proteins with varying activities as a defense mechanism to enhance survival in host cells and promote bacterial virulence. This usually occurs by enhancing attachment to eukaryotic cells, scavenging toxic ROS, or intoxicating target cells that assist in survival [1]. Cellular functions are highly regulated and consist of a series of interconnected signaling pathways that include kinases, phosphates, and other proteins [8]. Once an effector is successfully delivered into the host cell, it will attempt to mimic the endogenous activity of the host and manipulate signal transduction pathways that lead to ROS generation.

The role of the diet and its components in facilitating these processes is not well documented, particularly in terms of identifying specific food bioactives and related mechanisms that can influence the pathogen–host relationship. Polyphenolic compounds, for example, are consumed from the diet daily from plants in amounts approximating 1 g, thus representing a relatively large group of naturally diet derived antioxidants that are common to many fruits and vegetables as well as beverages such as coffee, tea, and wine [9]. Epidemiological studies have identified polyphenol rich foods to potentially reduce the risk of many chronic diseases [10]; however, there is a paucity of data to clearly show how they may inhibit bacterial infection. It is generally understood that an underlying mechanism for these capacities involves an antioxidant capacity to interact with host homeostatic pathways that control intracellular redox-balance. There are few studies to show that some phenolic compounds can affect the secretion system of pathogenic bacteria, which in turn influences virulence and facilitates pathogenicity.

## 2. Pathogen-Host Relationships

### 2.1. Bacterial Secretion Systems Facilitate Survival of Bacteria within the Host Cell 

Once the bacteria invade the host cell, they are exposed to a variety of killing mechanisms that include exposure to ROS generated by a respiratory burst response. Respiratory burst, also known as oxidative burst, is an enhanced release of ROS such as superoxide radicals and hydrogen peroxide by innate immune cells [11]. The innate immune system is activated and phagocytes are recruited to the infection site to eliminate the invading bacteria. The release of ROS signals a function to trigger the activation of the human immune system that involves the phagocytosis of bacteria and viruses. The phagocytosis of bacteria will activate host NADPH oxidase (NOX2) activity to produce hydrogen peroxide and superoxide radicals. The induction of nitric oxide synthase (iNOS) also occurs, which generates NO; this leads to a reaction with superoxide radicals that is produced by NADPH oxidase to form toxic reactive nitrogen species (RNS) such as peroxynitrite and nitrogen dioxide. During an infection, NADPH oxidase in immune cells reduces O_2_ to H_2_O_2_. Monocytes and neutrophils utilize myeloperoxidase to create hypochlorite via the combination of H_2_O_2_ with Cl^−^ to destroy bacteria, a reaction that is shown in Figure 1 [11,12].

For pathogens to survive the respiratory burst, specialized secretory mechanism(s) that include effectors to enter the host cell must be employed by bacteria. Effector proteins transferred from pathogens into the host cell occur through a variety of secretion mechanisms that include the type III secretion system (T3SS), type IV secretion system (T4SS), and type VI secretion system (T6SS). These systems have a needle-like machinery in common to inject the effector across a phospholipid-rich membrane [13]. The effector proteins are synthesized within the bacterium, but remain inactive due to lack of appropriate substrate and/or absence of host cell activators. Bacterial effectors have the capacity to inactivate hydrogen peroxide and in doing so will interrupt the production of toxic reactive species. This in turn aids in the persistence and survival of bacteria within the host cell. Table 1 summarizes some examples, demonstrating how bacteria respond to a host derived oxidative stress response.

A good example of this activity involves enterohemorrhagic *Escherichia coli* (EHEC), a pathogen which uses a secretion system to secrete a protein effector in response to oxidative stress that is induced by activated macrophages designed to protect against oxidative stress. Detoxifying enzymes that contribute to resistance to ROS in *E. coli* include several catalase types such as hydroperoxidase I (HPI), KatG and hydroperoxidase II (HPII), and KatE [14,15,16,17]. Studies have shown that bacteria growth can be achieved through the expression of a KatN gene, which produces catalase KatN proteins [18]. KatN is a catalase effector protein that decreases macrophage ROS generation, thus promoting the growth of intracellular EHEC. Once EHEC has been phagocytized by macrophages, the KatN gene is upregulated, leading to KatN secretion to hydrolyze host derived ROS that are generated in the cytoplasm [18].

The Gram-positive bacterium, *Enterococcus faecalis*, is a commensal organism affecting humans and also survives an oxidative burst derived from macrophages [19]. This organism secretes a catalase, namely KatA, which is a contributing factor to its resistance to host derived oxidative stress [7]. Three *E. faecalis* peroxidases have been identified to have critical roles in modifying stress resistance: NADH peroxidase (Npr), alkyl hydroperoxide reductase (Ahp), and thiol peroxidase (Tpx) [23].

There are additional mechanisms that reduce ROS exposure that are specific to other pathogens and produce a resistance to eventual cell death. For example, *Salmonella enterica* serovar typhimurium under oxidative stress will sequester manganese, an important cofactor responsible for the activation of Mn superoxide dismutase (MnSOD). This action promotes SodA and KatN enzyme activity [20], which are similar in function to catalase in initiating ROS degradation. The net effect of this activity is an increased survival of *Salmonella* due to decreased intracellular ROS concentration.

Other examples of catalase activity derived from pathogens include the KatB-LacZ and the KatB catalases that are present in *Pseudomonas aeruginosa* and are induced 250-fold when exposed to oxidative stress [24]. Similarly, upregulation of *katA* gene expression has been detected during the exponential growth phase in *Sinorhizobium meliloti,* when induced by exogenous H_2_O_2_ [22]. *S. meliloti* has all three catalase genes that will code for HPII, KatA, and KatC. Evidence also exists that the *katA* gene is regulated by OxyR, which also responds to oxidative stress [22].

### 2.2. Mechanisms for Microbial Effectors to Influence Host Cell Signaling

Cell signaling processes enable communication between cells in response to a change in the extracellular microenvironment where the redox balance ultimately regulates basic metabolic homeostatic processes such as cell growth and development [25]. An interaction that occurs between a specific food pathogen and host will potentially modulate the host redox balance, facilitated through a cell signaling mechanism [13,26,27]. Interference with normal cell signaling is a key approach for pathogen survival in the host [13]. Several studies have demonstrated that the role of pathogen virulence factors is to manipulate the host cell signaling pathway by producing pathogen specific effector proteins (Table 2). The virulence factors that attack kinase cascades are involved in intra- and extra-cellular signaling, where regulation of GTPase activity and modification of the cytoskeleton structure and function, as well as ubiquitin-dependent pathways, transform cell signal stability [28].

#### 2.2.1. MAPK (Mitogen-Activated Protein Kinases)

Mitogen-activated protein (MAP) kinases (MAPK) are a family of kinases activated through a phosphorylation cascade that consists of MAPKKK (RAF), MAPKK (MEK), and MAPK (ERK 1/2); this collectively is also often referred to as the MAP kinase module [29]. Three variations including the extracellular signal-regulated kinases (ERK), c-jun N-terminal kinases (JNKs), and p38 MAPK are categorized based on the specific TxY sequence (at threonine and tyrosine residues, respectively) that are phosphorylated during the activation of the MAP kinase module [29]. ERKs are directed mostly to cell proliferation and differentiation processes, while JNK and p38 MAP kinases regulate cellular responses that monitor cellular stresses such as inflammation [8].

There is strong evidence to show that a type III effector protein derived from *Shigella*, referred to as OspF, can inhibit JNK, ERK1/2, and p38, the three main families of mammalian MAPKs [30]. Inactivation of these kinases prevent the phosphorylation of serine residues in histone-3, which in turn is responsible for reducing kinase activation and thus effectively inhibits the MAPK pathway [31]. *Shigella* can also deliver other effector proteins such as OspB, which is required for full activation of p38 and ERK1/2 of the MAPK pathway [30]. It has been hypothesized that OspB activates the cytosolic phospholipase A2 (cPLA2), which facilitates secretion of chemo-attractants such as IL-8 [30]. Since OspF and OspB effectors have opposite effects on MAPKs, the presence of both enable *Shigella* the ability to control a host cell response during infection, which in turn leads to a higher probability of invasion. In the presence of OspB, the phosphorylation state of the three main families of MAPK (ERK1/2, p38, JNK1/2) has been observed to be relatively fast, showing activity within an hour. In the absence of OspB, a delay in the phosphorylation of MAPKs occurs at a rate that exceeds one hour. This suggests that OspB activity will enhance inflammation and have a role in the promotion of an inflammatory response at a particular stage of infection. Support for this conclusion comes from early stages of *Shigella* infection, where the pro-inflammatory effect of OspB to activate MAPK and polymorphonuclear (PMN) migration leads to bacterial internalization and proliferation [30]. Once internalized, *Shigella* utilizes the anti-inflammatory effect of OspF to prolong survival inside the host tissue. Further support for this comes from the observed increased movement of PMN leukocytes to the infection site, which increases the permeability of the epithelium barrier and facilitates the process whereby bacteria cross the basolateral cell membrane [32].

*Yersinia* is a pathogenic bacteria responsible for the plague known as the Black Death. This pathogen uses type III effector proteins to modulate cellular signaling pathways such as MAPK and the nuclear factor κB (NF-κB) during an innate immune response [33]. The effector protein in this organism is termed YopJ, and has acetyltransferase activity that utilizes coenzyme A to modify both serine and threonine residues that are critical for the phosphorylation of MAPKK and IKK-beta [34]. This action blocks downstream phosphorylation and prevents the activation of the protein cascade for this pathway. The acetylation could occur in a two-step process, with the YopJ acetylated on Cys-residue via a thioester bond [33]. This is very likely due to the similarity between YopJ and cysteine proteases. The second step of the reaction is the nucleophilic attack of a hydroxyl moiety on the thioester bond forming an acetylated amino acid [33]. The location of the acetylation directly competes with downstream modification and phosphorylation, thus inhibiting the pathway completely [33].

Using the type three secretion system (T3SS), the organism *Edwardsiella piscicida* can translocate EseK into the host cell and inhibit the phosphorylation of p38, JNK, and ERK1/2. This function leads to alleviating MAPK-mediated immune responses and will contribute to bacterial infection [35].

#### 2.2.2. Phosphoinositide 3-Kinase (PI-3 Kinase)

Pl-3 kinase are highly regulated and activate various signaling pathways that control for cell cytoskeletal dynamics and migration, cell growth, survival, death, and proliferation [36,37,38]. The PI3K pathway can be triggered by activated receptor kinases or ligand binding to integrins, leading to activation of focal adhesion kinase (FAK) or integrin-linked kinase (ILK). These kinases recruit adapter protein Crk and p130 Cas as well as tyrosine kinase Src. Thereafter, phosphorylated Src can contribute to the association of PI3K with the signaling complex (Figure 1) [6]. Studies have shown that EPEC delivers effector proteins Map, EspF, Tir, and Intimin into macrophages that inhibit phosphatidylinositol-3 (PI-3) kinase-dependent pathway activation [39]. EPEC and other Gram-negative bacteria utilize a T3SS system to secrete effector proteins into host cells. Further studies have shown that EspF is linked to mitochondria targeting and will contribute to the disruption of the cell epithelial barrier [39]. Research has shown that EspF inhibits Pl-3 kinase and functions independently from the mitochondria of the cell [39] to ultimately disrupt the epithelial barrier, which facilitates bacterial uptake [39]. *Yersinia* spp. also uses the type III secretion system to secrete a tyrosine phosphatase YopH [40,41]. YopH has been shown to bind to p130Cas and FAK, thus disrupting focal adhesion complexes and inhibiting phagocytosis. As a consequence, bacteria internalization will be limited [42]. Effector protein SigD secreted by *Salmonella* typhimurium has been shown to activate Akt signaling in epithelial cells [43]. Studies have also shown that *Salmonella* typhimurium could use the type III secretion system to secrete inositol phosphates SopB to mediate sustained activation of Akt, thus affecting the PI3K pathway [44].

#### 2.2.3. Nuclear Factor Kappa-Light-Chain-Enhancer of Activated B Cells (NF-κB)

ROS modulate NF-κB activation both in a positive and negative manner [45,46] with the duration of the exposure to oxidative stress being a critical factor. Wu and others (2009) used human lens epithelial cells (HLECs) as a model to show that exposure to ROS at physiologically relevant levels caused I-κB degradation and subsequent NF-κB activation. However, sustained oxidative stress in cells will also lead to oxidative inactivation of the proteasome, thereby attenuating I-κB degradation and subsequent NF-κB activation. The activation of NF-κB is indispensable for HLEC to recover from oxidative stress [47]. Thus, the NF-κB pathway can be activated at a low level of oxidative stress, whereas with higher oxidative stress the proteasome is involved in the inactivation of signaling [47]. Impairment of the NF-κB signaling pathway will contribute to cell death, therefore it is critical to maintain ROS generation at physiologically relevant levels required for optimal redox biology.

Using dendritic cells, *E. coli* has been shown to utilize the effector protein NleE to inhibit the activation of NF-κB and cytokine production [34]. Of interest is the finding that enteropathogenic *Escherichia coli* (EPEC) utilizes different secretion systems to colonize in the gut. With three secretion systems, the EPEC injects effector proteins into epithelial cells to disable immune system signaling [34]. Mutant EPEC, developed to be unable to inject effector proteins, has displayed NF-κB activation that subsequently triggers the secretion of cytokines and pro-inflammatory mediators [34]. This result was supported by the finding that a deletion in a NleE gene resulted in the rapid excretion of cytokines. In contrast, complementation of NleE strongly reduced the production of cytokines. Therefore, NleE effector protein activity is a key factor to the inhibition of NF-κB [34].

*Salmonella* Typhimurium secretes an effector protein, AvrA, that can regulate host inflammatory response through β-catenin and NF-κB pathways, which results in deactivation of host inflammatory responses [48]. The deubiquitinase activity has been characterized, and the results show that AvrA can inhibit degradation of β-catenin and IκBα in epithelial cells. This activity leads to the inhibition of NF-κB activation. Interleukin-6 target genes of the NF-κB pathway are also downregulated.

The *Escherichia coli* virulence protein NleH1 has a capacity to prevent the nuclear translocation of ribosomal protein S3 (RPS3) [49]. RPS3 is nuclear translocated and functions parallel to p65 [50]. Both p65 and RPS3 bind to form the NF-κB complex, which associates with the regulatory site to promote transcription [50]. Inhibition of nuclear translocation of ribosomal protein subsequently inhibits the formation of NF-κB complex in *E. coli*.

#### 2.2.4. p21 Activated Protein Kinase (PAK)

The serine/threonine kinase p21-activated kinase (PAK) has an important role in regulating diverse cellular processes in most eukaryotic systems that include cell survival, cell cycle progression, scaffold function, and structure of cytoskeleton [51,52,53,54]. The fact that PAK has a critical role in the functionality of host cells makes it very interesting to note that it is also a target for most pathogens [55]. EHEC, for example, uses its secretion system to secrete EspG to interact with the Rac/cell division cycle (Cdc42) (p21) binding domain of PAK [56,57]. *Salmonella* typhimurium uses its type III protein secretion system to secrete effector proteins to activate Cdc42 and the PAK1 signaling, supported by evidence showing that NF-κB signaling induced by *Salmonella* typhimurium in cultured cells is inhibited after the removal of Cdc42, PAK1, TRAF6, or TAK1 [58]. *Helicobacter pylori* uses a type IV secretion system to translocate CagA protein into gastric epithelial cells, and CagA protein undergoes tyrosine phosphorylation, which is involved in stimulating the signaling pathways including small Rho GTPases Rac1 and Cdc42 and activation of PAK1 [59]. In addition, phosphorylated CagA protein results in the dephosphorylation of α-Pix, which is a PAK-binding protein that may be involved in the activation of PAK. Thereafter, dephosphorylated α-Pix is capable of regulating PAK, which is responsible for modulating cytoskeletal changes in gastric epithelial cells [60]. *P. aeruginosa* uses the type III secretion system to secrete ExoS, ExoT, ExoU, and ExoY effectors into host cells, leading to actin cytoskeleton rearrangement of host cells to favor their subsequent internalization [61].

## 3. Disruption of Pathogen–Host Cell Interaction and Cell Signaling Pathways by Polyphenols

There are relatively few studies that have demonstrated the relevance of consuming dietary polyphenols on pathogen–host cell interactions [75]. Dietary polyphenols including flavonoids and phenolic acids derived from numerous fruits, vegetables, and beverages such as tea and coffee have well defined antioxidant activities that relate to a capacity to directly scavenge free radicals, hence preventing ROS propagation [76,77,78]. On the other hand, some antioxidants also exhibit prooxidant activity under certain conditions [79], which is related to both the concentration and the collective mixture of antioxidant and prooxidant activities present in the reaction milieu.

### 3.1. Effect of Polyphenol on Adhesion of Host Cells

The adhesion of pathogens to host cells is a critical prerequisite for bacteria colonization and the initiation of a majority of infectious diseases that will ultimately lead to a bacteria-derived pathogenesis [80,81]. Colonization, followed by internalization, contributes to the delivery of toxin(s) and virulence factors to host cells as well as aiding in a bacteria defense mechanism against host immunity [81]. Many Gram-negative bacteria inject effector proteins into host cells using secretion systems that include type III, type IV, and type VI systems [82].

Polyphenolic compounds, some derived from dietary sources, have been shown to protect against intestinal barrier tight junction proteins exposed to oxidative stress [83] and also inhibit adhesion of some types of pathogenic bacteria to host cells [84,85,86] as well as to display anti-adherence effects on specific bacteria types [87,88]. For example, cranberry extracts have an affinity to inhibit the adherence of *H. pylori* in the gastrointestinal tract [89] and *Streptococcus mutans* in the oral cavity [90].

The polyphenolic fractions extracted and subsequently concentrated from bilberry, cranberry, crowberry, and lingonberry juice have exhibited anti-adhesive properties that function against *Neisseria meningitidis* and HEC-1B human epithelial cells [86]. The *N. meningitidis* actually binds to polyphenolic fractions of berry juices, thus protecting the host against infection [86]. Some specific examples of polyphenols derived from cranberries have been shown to reduce the adhesion of *E. coli* [85] and *S. mutans* [84]. Polyphenols have been shown to affect the adhesion of representative gut flora to cultured Caco-2 cells in vitro, and potentially alter the gut microecology [91].

### 3.2. Mechanisms by which Dietary Polyphenols Manipulate Host Redox Environment

The antioxidant activity of polyphenols has been ascribed to a capacity to scavenge free radicals and also sequester reactive metal ions; the latter reduces the initiation rate of oxidation. Free radical scavenging activities of many polyphenols include quenching hydroxyl radicals (•OH) [92], superoxide anion radicals (O_2_^•−^) [93,94], and the stable, non-reactive 1,1-diphenyl-2-picrylhydrazyl (DPPH) radical [93,95] (Table 3). Another cell-based mechanism involves the tripeptide glutathione (GSH), an abundant non-protein thiol metabolite that has an immensely important role in mediating a wide range of redox reactions by cycling between dithiol-reduced and disulfide-oxidized forms. Polyphenols have relevance to cytosol and mitochondrial cell redox homeostasis through a capacity to facilitate GSH scavenging of ROS through a polyphenol-ascorbate-glutathione cycle. GSH also reacts with NO to form GSNO, which has a key role in inducing defense genes that act to modulate tolerance to oxidative stress. Polyphenols also interact with metal ions through aromatic hydroxyl groups, which function as a sequestering agent [96,97,98,99,100,101], thereby lowering the activity of the transition metal ion to initiate peroxidation [102].

Dietary polyphenols also display pro-oxidant activity under specific conditions that has been ascribed to the generation of ROS such as •OH, O_2_^•−^, and hydrogen peroxide (H_2_O_2_), respectively, by the autoxidation of polyphenols [113,114,115]. Several studies have found that catechins, derived from tea, for example, when added to cell culture media or buffer solutions will lead to a significant H_2_O_2_ production [94,116,117,118]. In the presence of Fe^2+^, generation of the •OH radical would also result from the Fenton reaction. The generation of H_2_O_2_ in these reactions accounts for the variable effects observed from polyphenols on cells under different cell culture conditions. Moreover, studies have also demonstrated that considerable concentrations of H_2_O_2_ can be produced from polyphenol-rich beverages such as cocoa, green tea, black tea, and coffee during processing [119]. These findings have led to the observation that induction of apoptosis and mutagenicity can be eliminated in vitro when catalase is added to the reaction mixture [94,120,121]. Experiments using healthy volunteers have furthered this concept by demonstrating that typical coffee drinking practices will increase urinary H_2_O_2_ levels, hence advocating that coffee drinking could represent a dietary factor regulating human redox potential in a positive manner that would reduce oxidative stress [119].

As noted above, a critical component of pathogen survival in the host is the redox state of the microenvironment from which the pathogen comes into contact with the host. Neutrophil superoxide dismutase and myeloperoxidase are two enzymes involved in oxygen dependent killing mechanisms that involve *S. aureus* [122]. Furthermore, studies with *S. aureus* have shown that catalase activity is low during the initial exponential or log phase of bacterial growth, but increases steadily during the stationary phase [122]. For example, there is a resistance to *S. aureus* from peritoneal macrophage during its log phase. The intracellular survival of *S. aureus* after phagocytosis results in recurrent infections and a failure of antibiotic treatment, if the antibiotic has poor phagocyte penetration.

The ability of a pathogen to exploit the host’s inflammatory response starts for most pathogens with a virulence mechanism. Since modulation of the inflammatory response is sufficient to alter morbidity induced by the pathogen, the ability to inactivate H_2_O_2_ by the pathogen is related to its affinity to resist the pro-oxidative stress environment induced by the host in response to lower pathogen survival [123]. One mechanism where this can be achieved involves the H_2_O_2_-mediated oxidation of cysteine residues within proteins that contribute to redox signaling [124]. Upon recognition of chemical signal(s) that are attributed to oxidative and electrophilic molecules, cysteine residues are modified. At a physiological pH, cysteine residues exist in the thiolate anion (Cys-S^−^) form, which is susceptible to oxidation relative to that of the corresponding protonated cysteine thiol (Cys-SH) molecule [125]. For example, during redox signaling, H_2_O_2_ oxidizes the Cys-S^−^ to the sulfenic form (Cys-SOH), which then leads to allosteric changes within the protein, which in turn alters protein functionality. The Cys-SOH can be reduced to Cys-S^−^ by disulfide reductases, thioredoxin (Trx), and glutaredoxin (Grx) to return to original protein function [126]. Thus, oxidation of cysteine residues to Cys-S^−^ within proteins is considered to be a reversible mechanism of signal transduction [127]. The thiolate oxidation product reacts in living cells at nanomolar concentrations of H_2_O_2_, whereas a high level of H_2_O_2_ oxidizes thiolate anions to sulfinic (SO_2_H) or sulfonic (SO_3_H) species. In contrast to sulfenic modifications, sulfinic and sulfonic modifications are irreversible and cause permanent protein damage. Cells that have enzymatic mechanisms to overcome the build-up of intracellular H_2_O_2_ such as possessing peroxiredoxins and glutathione peroxidase [127] activities can resist this reaction. Cysteine, which is a critical component of NF-κB activation [128], may also be an important link between the onset of oxidative stress and induction of inflammation. Since some dietary polyphenols have a potential for both antioxidant and pro-oxidant activity, the conditions for these phytochemicals to specifically modulate the host redox environment is an important event when defining the conditions of altering pathogen–host interactions that are involved in promoting host resistance.

### 3.3. Targeting Virulence of Pathogen by Manipulating Phagocytosis

Polyphenols that alter phagocyte function by disrupting phagocyte and intracellular cell signaling pathways can influence bacterial pathogenesis [129]. One example of this is with the phytochemical remedy, pycnogenol, a water soluble mixture of catechins derived from grape skin or pine bark with noted antioxidant activity and a capacity to modulate immune responses [130,131]. Similar polyphenol compounds extracted from green tea can upregulate cathepsin D expression in RAW 264.7 cells [104] and inhibit Mycobacterium tuberculosis survival in human macrophages [132]. Moreover, epigallocatechin-3-gallate, a potent catechin recovered from tea, can downregulate tryptophan-aspartate containing coat protein (TACO) gene transcription in human macrophages. This is accompanied by inhibition of mycobacterium survival within macrophages, a likely ROS-derived mechanism for epigallocatechin-3-gallate in the prevention of tuberculosis infection.

### 3.4. Effect of Polyphenol on Bacterial Effector Secretion

Typically, the use of antibiotics is the most widely used method to control against bacterial infection; however, the generation of antibiotic-resistant bacteria has become a risk factor when practices of extensive and prolonged use are followed. Rather than focusing on bacterial viability, targeting specific virulence factors through mechanisms that involve secretion of bacterial effectors is an emerging strategy to lower bacteria pathogenesis without the risk of inducing bacterial resistance. Bioactive phytochemicals derived from the diet that have this capacity could be potentially important chemoprotective agents that act against pathogenic virulence. Examples exist with *P. aeruginosa*, where resveratrol tetramer (-)-hopeaphenol present in grape skins was shown to inhibit virulence factors in *P. aeruginosa* and *Yersinia pseudotuberculosis* [106,107,108]. The type III secretion system is an important virulence factor for *P. aeruginosa* that functions to secrete and translocate T3 effector proteins into human host cells. A more detail analysis of the mechanism has plant phenolics altering *exoS* transcription, which encodes a T3-secreted toxin by affecting the expression levels of the regulatory small RNAs, namely RsmY and RsmZ, through the GacSA-RsmYZ-RsmA-ExsA regulatory pathway [133]. Another example exists with *p*-coumaric acid (PCA), a phenolic compound and precursor compound to more complex polyphenols present in many fruits and vegetables, with a capacity to repress the expression of T3SS genes present in plant pathogen *Dickeya dadantii*, through a HrpX/Y two-component system. A structure–function relationship that involves both the hydroxyl group and the double bond of PCA has been linked to its activity [134]. The polyphenol (−)-epicatechin gallate, common to tea, has been shown to suppress the secretion of coagulase and alpha-toxin by *Staphylococcus aureus* by modifying the morphology and beta-lactam antibiotic susceptibility as well as the cell wall architecture of *Staphylococcus aureus.* Plant phenolic compounds 4-methoxy-cinnamic acid (TMCA) and benzoic acid (BA) have been shown to suppress T3SS of *Erwinia amylovora* through the HrpS-HrpL pathway. In addition, the *rsmB_Ea_*-RsmA_Ea_ system also contributes to the inhibition activity of TMCA against the T3SS system [135]. The extent that these activities are initiated by the change in redox balance of the host remains to be determined, but it offers exciting opportunities for potential new discoveries in this area. Other examples with plant pathogens include *Xanthomonas oryzae* pv. *oryzae* (*Xoo*) and *X. oryzae* pv. *oryzicola* (*Xoc*), two well-known bacterial pathogens that involve rice staples. Studies show that phenolic compounds such as *o*-coumaric acid can attenuate the virulence of these pathogens by targeting for T3SS [136]. Other studies have shown that plant-derived compounds are able to inhibit, or induce, the T3SS in plant pathogenic bacteria *Ralstonia solanacearum* through the HrpG-HrpB pathway [137].

### 3.5. Effect of Polyphenol on Host Cell Signaling

Polyphenolic compounds are found to interfere with membrane intracellular receptors as well as modulate important signaling mechanisms. Table 4 summarizes some examples on the effect that specific polyphenols have on cellular pathway mechanisms. Since both the bacterial effector and the polyphenol can affect host cell signaling, respectively, it is worth looking at which one has the predominant role in the hosts.

## 4. Conclusions

When a pathogen invades a host, it is critical that it exhibits the ability to modify the host immune response along with other defense systems in order to survive and multiply. Pathogens encode various effectors that have important roles in terms of the host cell response to the pathogen. These virulence factors are multifunctional and can assist the pathogen to invade the immune system by deactivating cellular pathways and downregulating host cell immune responses. Apart from escaping from the host immune response, some pathogens also utilize intracellular and extracellular host kinases and interrupt cellular pathways that are germane to its own life cycle. Since cellular kinases are key regulators in these pathways, once they are controlled, they can enable pathogens to express activity toward adhesion, cellular apoptosis, and modified cytoskeleton structures. Therefore, understanding how pathogens target cell signaling during the initial invasion process in order to modify host redox biology to promote replication will be important for developing new strategies that are effective at decreasing bacterial pathogenicity.

## Figures and Tables

**Figure 1 ijms-20-06222-f001:**
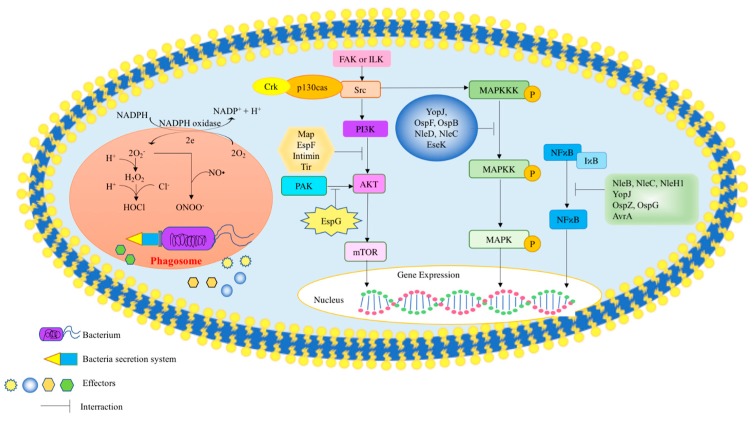
A scheme to show the effect of bacterial effectors on cell signaling. During phagocytosis, the host generates ROS to kill invading bacteria. To overcome the oxidative stress, bacteria produce effectors that facilitate survival in the host by directly scavenging ROS or indirectly interrupting cell signaling mechanisms that ultimately changes the host redox balance.

**Table 1 ijms-20-06222-t001:** Effectors secreted by bacteria against oxidative stress.

Bacteria	Effector	Reported Action	Reference
*EHEC*	Catalase KatN	Hydrolyze ROS	[18]
*Enterococcus faecalis*	KatA	Survive oxidative burst	[19]
*Salmonella spp.*	SodA and KatN	Detoxify ROS	[20]
*Pseudomonas aeruginosa*	Catalase KatB-AnkB, AhpB, and AhpC-AhpF	OxyR-dependent regulation against oxidative stress	[21]
*Sinorhizobium meliloti*	Catalase HPII, KatA and KatC	Decrease ROS concentration	[22]

**Table 2 ijms-20-06222-t002:** A summary of different bacterial effectors and the mechanisms involved in the interruption of host cell signaling.

Bacteria	Effectors	Cell Signaling Mechanisms	References
*E. coli*	Type III effectors NleE and NleB protein	Down-regulate NF-κB by decreasing IKK phosphorylation	[34,62]
	NleD (metalloprotease)	Inactivate JNK and p38.	[63]
	NleC (zinc protease)	Suppress inflammatory response by inactivating NF-κB and p38	[64,65]
	Tir	Suppress both TRAF2 and TRAF6-induced NF-κB activation	[66,67]
	NleH1	Suppress RPS3-induced NF-κB activation	[49,68]
	NleH2	Suppress NleH1	[68]
	EspH	Modulator of host cell actin cytoskeleton	[69]
	Locus of enterocyte effacement (LEE) encoded Map, EspF, Tir and Intimin proteins	Inhibit phosphorylation of Pl-3 kinase substrate	[39]
*EHEC*	Stx2 toxin	Pro-inflammatory protein, promote IL-8 production.	[70]
	EspG	Inhibit PAK signaling	[56,57]
*Yersinia*	YopJ	Anti-inflammatory activity that inhibit both MAPK and NF-κB pathways	[33]
	YopH	Inhibit Ca signaling and ROS production	[71]
	YopE (GTPase-activating protein)	Downregulate Rho, Rac and Cdc42 activity	[72]
*Shigella spp.*	OspF (Phosphothreonine lyase)	Inactivate MAPK	[73]
	OspB (Induces phosphorylation)	Activate MAPK	[30]
	OspZ	Inhibit NF-κB	[62]
	OspG	Stabilize IkB	[74]
*Salmonella spp.*	AvrA (deubiquitinase)	Remove ubiquitin from IκB-alpha and beta-catenin	[48]
*E. piscicida*	EseK	Inhibit MAPK	[35]

**Table 3 ijms-20-06222-t003:** Free radical scavenging activities of polyphenols.

Phenolic Compounds	Free Radical Scavenging Activity	Reference
Protocatechuic acid	Best against DPPH• and O_2_^•−^	[103]
Pyrogallol	Best against DPPH• and O_2_^•−^, effective against ABTS, DMPD, H_2_O_2_	[104]
Caffeic acid	Best against DPPH• and O_2_^•−^	[105]
Gallic acid	Best against DPPH• and O_2_^•−^	[106]
Sinapinic acid	Hydroxyl radical scavenging	[107]
Chlorogenic acid	Hydroxyl radical and O_2_^•−^	[108,109]
Epicatechin	DPPH• scavenging, hydroxyl radical, and superoxide anion radical-scavenging activities	[110]
Naringenin	Hydroxyl and superoxide radical scavenger	[111]
Luteoloside	Against H_2_O_2_ radicals	[112]
Apigenin	Against H_2_O_2_ radicals and DPPH• scavenging	[112]

**Table 4 ijms-20-06222-t004:** Polyphenols capacity to interact with the cellular pathway and the mechanism behind it.

Polyphenols	Food Sources	Signaling Pathway	Mechanism	References
Catechin, theaflavin, and thearubigin	Tea products	Membrane intracellular receptors	Activation of cellular receptors which modifies intracellular signaling. Directly bind to ribonucleic acid (RNA) and deoxyribonucleic acid (DNA), allowing them to induce gene transcription. Assist in nuclear translocation of other transcription factors. Interact with transcription activator or repressor in the nucleus, altering gene expression.	[138]
Curcumin	Ginger	Interferes protein kinase C (PKC) signaling pathway, oncogenes and coding proteins	Regulates the transcription of the antioxidant enzyme genes through PKC signaling. Curcumin suppresses the activity of PKC which indirectly affecting MAPK and C-jun.	[139]
Equol, kaemferol, resveratrol, ellagic acid	Found in vegetables such as spinach, kale and endive.	Tyrosine kinase	Inhibits HCC827 panel, tyrosine kinase inhibitor (TKI)-sensitive (TKIS) and TKI-resistant clones.	[140]
Gallic acid, p-courmaric, heaperidin	Bark, wood, leaf, fruit, root and seed. Present in berries, plums, grapes, mango, tea, wine	Tyrosine kinase	Inhibit only tyrosine kinase inhibitor (TKI)-resistant TKIR cells H1993.	[140]
Genistein	Soy-based products such as chickpeas, tofu, soymilk, soy flour, soy protein, miso, tempeh	DNA methylation and histone modification.	Decreases DNA methylation of various tumor suppressor genes. Demethylate and reactivate TNF-stimulated gene (TSG), causing anticancer effect. DNA methylation is a critical part of transcriptional regulation that is catalyzed by specific DNA methyltransferases (DNMTs). Genistein alters histone to promote or prevent DNA replication.	[141]
Green Tea polyphenol (-)-epicatechin-3-gallate (ECG)	Green Tea	Activation of ERK and p38	Induced AU-rich elements (ARE)-mediated gene expression to activate MAPK pathway, stimulate caspase-3 activity and induce apoptosis Anti-cancer and tumor property comes from their ability to suppress cellular growth and initiate apoptosis.	[142]
Phytoestrogens	Soy products and foods such as soybeans, tofu, miso, tempeh, vegetables, fruits, grains and legumes.	Target classical estrogen receptors (ER) pathway, TNF signaling pathway, and non-genomic signaling	Binds to ERα or ERβ, and induces estrogen receptor expression (ERE)-dependent transcription. Inhibiting tumor necrosis factor-α (TNF-α)-induced apoptosis. Involvement in pathways allow phenolic compounds to cross-talk to other transduction signals and a wide application potential. Activates estrogen pathway to regulate the expression gene responsible for the maintenance of bone mass. Thus phytoestrogens help balance bone resorption and bone formation.	[143]

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
