# Peer review of "Molecular Mechanisms That Define Redox Balance Function in Pathogen-Host Interactions—Is There a Role for Dietary Bioactive Polyphenols?"

_ijms, 2019, doi:10.3390/ijms20246222_

Round 1

Reviewer 1 Report

The paper of Mu et al reviews important insights regarding the molecular mechanism of redox balance in host-pathogen interaction. The paper is well written, well documented and moreover I appreciated the chapter related to diet polyphenols. I suggest to authors to draw  a graphical abstract that links the aims of this review.

Regarding chapter of pathogen -host relationship, please cite

https://doi.org/10.1002/pmic.201400276

Author Response

Please find our edited version of the manuscript. We went through the paper and made number of changes that reviewers commented on. We have drawn a graphical abstract that links the aims of this review. 

Reviewer 2 Report

This review article examines the mechanism and interactions surrounding ROS production and signalling with respect to bacterial pathogens and the presence of polyphenols. This is a well written and interesting review. I only have minor comments (below) and my only major point is that I do not feel the title is suitable. Please change the title to bacterial host interactions or include detail of fungi, parasites and viruses and the subject of bioactive phytochemicals needs to be included in the title and the abstract as this is a key part of the review.

'To ensure a functional immune system, the host must detect and respond to the presence of pathogenic bacteria during mammalian infection'.

And fungi, parasites and viruses. Perhaps change 'bacteria' to 'microbe' anywhere it is not microbe type specific? Check carefully as this is a problem throughout.

l49-51. This paragraph seems a big jump in subject. How does this link with that which comes before? There important differences in the location of ROS production between e.g. neautrophils and macrophages, and between different organelles e.g. production at the plasma membrane, secreted granules, endosome/phagosomes, cytoplasmic and mitochondria. These details are vital in understanding the subject in general and the interaction with microbial mechanisms. l103 serovar should not be in italics. Change the title of section 3. It does not suitably explain what this section is about; it is all about polyphenols.

Author Response

Thank you very much for your comments. Please find our revised paper, entitled “Molecular mechanisms that define the role of redox balance in pathogen-host interactions. Is there a role for dietary bioactive polyphenols.

We went through the paper and made number of changes that reviewers commented on. We have drawn a graphical abstract that links the aims of this review. Please find attached the slides. We made some changes on the title and included the dietary phenolics in our title. We also made some changes on Table 3. We have change the title of section 3, and correct the italics in line 103.